# Lipid Nanovesicles for Antioxidant Delivery in Skin: Liposomes, Ufasomes, Ethosomes, and Niosomes

**DOI:** 10.3390/antiox13121516

**Published:** 2024-12-12

**Authors:** Agnese Ricci, Luca Stefanuto, Tecla Gasperi, Fabio Bruni, Daniela Tofani

**Affiliations:** 1Department of Science, Section of Nanoscience and Nanotechnologies, “Roma Tre” University, Via della Vasca Navale 79, 00146 Rome, Italy; agnese.ricci@uniroma3.it (A.R.); luca.stefanuto@uniroma3.it (L.S.); tecla.gasperi@uniroma3.it (T.G.); 2Department of Science, Section of Nanoscience and Nanotechnologies, “Roma Tre” University, Via della Vasca Navale 84, 00146 Rome, Italy; fabio.bruni@uniroma3.it

**Keywords:** antioxidants, liposomes, ethosomes, niosomes, delivery systems

## Abstract

The skin, being the largest organ of the human body, serves as the primary barrier against external insults, including UV radiation, pollutants, and microbial pathogens. However, prolonged exposure to these environmental stressors can lead to the generation of reactive oxygen species (ROS), causing oxidative stress, inflammation, and ultimately, skin aging and diseases. Antioxidants play a crucial role in neutralizing ROS and preserving skin health by preventing oxidative damage. In recent years, nanotechnology has emerged as a powerful tool for enhancing the delivery of antioxidants onto the skin. In particular, liposomal formulations have offered unique advantages such as improved stability, controlled release, and enhanced penetration through the skin barrier. This has led to a surge in research focused on developing liposomal-based antioxidant delivery systems tailored for skin health applications. Through a comprehensive analysis of the literature from the 2019–2024 period, this review provides an overview of emerging trends in the use of liposomal delivery systems developed for antioxidants aimed at improving skin health. It explores the latest advancements in liposomal formulation strategies, vesicle characterization, and their applications in delivering antioxidants to combat oxidative stress-induced skin damage and other associated skin pathologies. A comparison of various delivery systems is conducted for the most common antioxidants. Finally, a brief analysis of lipid nanovesicles used in the cosmeceutical industry is provided.

## 1. Introduction

Skin cells play a crucial role as the body’s first line of defense against external stressors. These stressors, including UV rays, pollution, and various infections, can increase the production of reactive oxygen species (ROS) such as O_2_^•^, HOO^•^, H_2_O_2_, and HO^•^. These species are naturally produced as byproducts in cell metabolism and are involved in redox signaling [1]. Cells are able to regulate ROS concentration using antioxidants (vitamin E, and C, glutathione, coenzyme Q10, carotenoids) and enzymes (superoxide dismutase, catalases, glutathione peroxidase, etc.) that help maintain redox homeostasis [2,3,4]. However, this regulatory capacity declines with age [5], leading to an imbalance in redox equilibrium, which results in oxidative stress and inflammation. In the skin, chronic oxidative stress can contribute to the development of wrinkles, loss of elasticity, and hyperpigmentation as well as to various pathological conditions such as psoriasis, dermatitis, and even cancer [6,7].

It is well known that the use of exogenous antioxidants supplied by the diet enhances the body’s response against ROS [2,8]. However, many natural antioxidants are not as effective in vivo due to their short lifetime, low absorption through cell membranes, rapid metabolic degradation, and excretion [9]. Topical application to the skin suffers from similar disadvantages, because many natural antioxidants are unable to penetrate the stratum corneum (SC). The structure and composition of the skin layer are shown in Figure 1.

In recent decades, several nanomaterials have been developed, gaining significant attention for their applications across various fields. Notably, extensive research has also been conducted for delivering molecules with poor solubility [10]. Some examples of nano-delivery systems include vesicles [11,12], liposomes [13], polymer nanoparticles [14], solid lipid nanoparticles [15], and liquid crystals [16]. These systems are designed to encapsulate and deliver drugs or other active compounds to specific target sites in the body, offering improved efficacy, reduced side effects, and enhanced therapeutic outcomes. On the other hand, lipid vesicles experience instability over time, limited active ingredient loading capacity, challenges in retaining the active ingredient, and high costs. Each system has its own unique properties, advantages, and drawbacks, making it suitable for different applications. In the case of skincare, specialized liposomal carriers have been tailored to target the unique characteristics of the skin’s external surface and to pass through the specific structure of the stratum corneum.

This review aims to provide a synopsis of various liposomal systems developed for delivering antioxidants into skin cells. It offers a general description of the critical quality attributes (vesicle size, size distribution, stability), the potential benefits and limitations of each type of vesicle, along with recent applications for delivering specific antioxidants to the skin and the use of lipid nanovesicles in the cosmeceutical industry.

## 2. Natural and Synthetic Vesicles

A natural vesicle is a small, membrane-bound cavity. Liposome vesicles are formed from the phospholipids of cell membranes and can vary in size from nanometers to micrometers. These sacs play crucial roles in transporting, storing, and processing various substances within the cells. Since Bangham’s discovery of the possibility of producing synthetic liposomes in the laboratory [14], their use as drug delivery systems has expanded significantly. Initially, liposomes were formed only from natural phospholipids (PL), such as dipalmitoyl phosphatidylcholine (lecithin or DPPC), phosphatidylethanolamine (cephalin), and phosphatidylserine derived from soybeans, eggs, and milk. Later, liposomes have also been produced using synthetic or semisynthetic molecules.

More recent formulations have envisaged the use of organic solvents (ethosomes) or surfactants (transethosomes) to enhance the vesicle deformability. On the other hand, the use of polyethylene glycol bound to the membrane to produce stealth liposomes allows for evasion of immune system recognition. Additionally, specific proteins can be added to the outer layer to enhance cell specificity and facilitate phagocytosis (actively targeted liposomes) [17].

The administration of liposomal formulations can occur via intravenous or lumbar injection or through topical administration [18]. Their biocompatibility and biodegradability have made them a popular choice in nanomedicine [19,20]. This has proven extremely useful in anticancer therapies with common drugs [21] and is paving the way for the pharmacological application of natural bioactive molecules that were previously overlooked due to their low solubility and poor pharmacokinetics [22]. Many natural antioxidants face similar issues and are seldom used, despite being highly effective against inflammation and diseases associated with ROS imbalance. Liposomes and the other lipid-based vesicles specifically designed for antioxidant delivery to skin cells allow for direct application on the epidermis, enhancing efficacy and absorption. This review traces the development of various lipid-based nanodelivery systems studied over the past five years, focusing specifically on skin treatments. The antioxidants examined are listed in Table 1.

### 2.1. Liposomes

Synthetic liposomes (LIP) can be produced with various methodologies [40,41,42]. The most commonly used methods include (i) thin film hydration; (ii) reverse phase evaporation; (iii) solvent injection; and (iv) micro hydrodynamic focusing [17]. The first method, used by Bargham, involves blending phospholipids and other lipophilic substances in an organic solvent. After gently evaporating the solvent, a thin phospholipid layer adheres to the walls of the flask. By slowly adding a buffered aqueous solution, vesicles are formed (Figure 2a). The reverse phase method initially produces inverted micelles by adding the organic phospholipid solution to the aqueous phase containing the antioxidant compound. The evaporation of both solvents facilitates the formation of unilamellar and multilamellar vesicles (Figure 2b). In the solvent injection method, the lipid fraction, diluted in an organic solvent, is injected into a boiling aqueous solution containing the active ingredient. The evaporation of the organic phase leads to the aggregation of the lipid bilayer (Figure 2c). A more recent system employs two thin channels with a diameter of less than 500 µm to allow the organic and aqueous solutions to flow. When the channels are brought into contact through a Y-shaped connector, the mixing of the phases produces small vesicles of similar diameter, with a low polydispersity index (Figure 2d).

LIP are composed of one or multiple bilayer membranes with variable diameters, made up of natural or synthetic phospholipids [43]. The choice of phospholipids used to prepare these vesicles significantly influences their physicochemical properties. Liposomes formed from saturated phospholipids exhibit a higher gel-to-liquid crystalline phase transition temperature, increased rigidity, and a greater ability to retain drugs within their bilayer. One potential limitation of these vesicles is their excessive stability, which can result in reduced release of the drug trapped inside.

On the contrary, LIP made from unsaturated phospholipids have properties that are the opposite. Therefore, the proper design of a vesicle nanocarrier involves a balanced mixture of phospholipids to achieve stability and effective drug release [44]. LIP can carry hydrophilic or hydrophobic molecules in their core or bilayer, making them versatile carriers that enhance drug efficacy while reducing side effects [45]. The efficiency of drug absorption can be quantified by calculating the encapsulation efficiency (EE), which represents the percentage of the drug that remains in the vesicles compared to the amount initially present in the solution. A schematic overview of various liposome types is shown in Figure 3.

Since the initial use of LIP as a drug delivery system, several drawbacks have been identified, and many efforts have been pursued to address them. To enhance bilayer stability against mechanical and chemical stress, additives such as cholesterol and vitamin E have been added. Cholesterol modulates membrane rigidity by reducing interactions between saturated chains and enhancing those between unsaturated ones [46]. More recently surfactants have also been used to balance membrane stability and deformability [29]. Additionally, because vesicles can aggregate, charged phospholipids have been preferred to maintain separation between vesicles through electrostatic repulsion [47]. The surface potential of vesicles is influenced by the choice of phospholipid head groups in the bilayer and the presence of a drug with a specific charge. The ζ potential (ZP), typically determined by microelectrophoresis, provides insight into this parameter. In particular, anionic LIP are easily phagocytized by macrophages and can penetrate cells via endocytosis, making these two aspects particularly useful in drug delivery through the skin [45]. The size of the vesicle is another important parameter to control during LIP preparation. Vesicles with a mean diameter (MD) in the range of 20–40 nm, referred to as small unilamellar vesicles (SUV, Figure 4), can evade immune system attacks, resulting in longer clearance times. In contrast, only vesicles smaller than 100 nm, known as medium unilamellar vesicles (MUV), can prolong their circulation in the blood and are more easily absorbed by cells [19]. Multilamellar vesicles (MLV) can also be produced. In the field of dermal drug delivery, even larger vesicles (LUV) can be used, provided they can penetrate the SC. It is important to note that the physicochemical properties of the entrapped organic compounds can affect the dimensions of the vesicles, especially when these molecules interact with the bilayer. In some cases, vesicle dimensions were increased due to the loosening of interactions between phospholipids from the intercalation of the drug molecule [24,29], while other drugs exhibited the opposite effect [13,31].

One important parameter for assessing vesicle size distribution is the polydispersity index (PDI). It characterizes the distribution of molecular mass among the vesicles in a sample, providing insight into the heterogeneity of their volume. Measured using Dynamic Light Scattering (DLS), a PDI value lower than 0.1–0.2 indicates a narrow size distribution and a more uniform population of liposomes. The uniformity is often desirable for consistent drug delivery and predictable behavior in biological systems. Techniques generally used to enhance liposomal size homogeneity include sonication during mixing operations [24] and filtration through filters smaller than the average diameter of the liposomes [26].

Many liposomal formulations have been used to deliver antioxidants, either pure or derived from complex natural extracts. Bavarsad et al. [23] developed liposomal formulations of pure quercetin for treating skin pressure ulcers. They employed the fusion method using soy DPPC (20% *w*/*w*), cholesterol (2% *w*/*w*), propylene glycol (6% *w*/*w*), and vitamin E (0.3% *w*/*w*), in the presence of quercetin buffer solutions at concentrations ranging from 0.01 to 0.04 g/mL, without employing organic solvents. The small unilamellar vesicles (SUV) produced exhibited a homogeneous particle size with a MD of 10–16 nm, demonstrated good encapsulation efficiency (EE 65–78%), and were sufficiently viscous for direct application to the affected areas. This delivery system demonstrated a high efficiency of retention (RE) into the skin (RE 46–51%) and a good penetration ratio (retention/penetration ratio of 7.1 to 9.0). This indicates that the majority of quercetin-loaded LIP were effectively retained in the skin. The active ingredient in the liposomes helped regulate vascular circulation and necrosis, reducing inflammation and shortening recovery time for pressure ulcers. Additionally, with minimal systemic absorption, the liposomal formulation exhibited fewer side effects compared to phenytoin cream, which was used as the reference drug. An example of research on plant extract delivery was explored through the study of *Moringa oleifera* leaves [13]. Rich in carotenoids, glucosinolates, vitamins, amino acids, and various phenolic antioxidants, *Moringa oleifera* is well known for its antimicrobial, anticholesterolemic, and anti-inflammatory properties. In the skin, this plant extract helps maintain collagen and hydration levels, thereby enhancing defense against aging and photodamage. The hydroethanolic extracts of the *Moringa oleifera* leaves were encapsulated in LIP formed with a 4:1 ratio of DPPC to cholesterol. The presence of the extracts reduced the diameter of the LIP from that of a giant unilamellar vesicle (GUV, 736 ± 100 nm MD) to that of a large unilamellar vesicle (LUV, 154 ± 1 nm MD). The loaded vesicles possessed a low polydispersity index (PDI 0.17 ± 0.01), a weak positive charge (ZP +6.6 ± 0.1), and a good retention rate (RR > 80% at 35 °C for 6 h). The antioxidant activity (AA), as determined by the DPPH assay, of the pure extract (AA 85.8 ± 1.5%) was similar to that of the reference, gallic acid, while the liposomal formulation showed lower activity (AA 70.0 ± 1.0%). Additionally, the antioxidant effect of liposomes, determined by ORAC experiments, diminished more rapidly (60 min instead of 120 min for the pure extract). A gel formulation containing propylene glycol, hydroxyethyl cellulose, and glycerin was used to analyze effects on skin. The results demonstrated that although the free hydroethanolic extracts exhibited greater antioxidant properties and enhanced hydration more effectively, the liposomal formulation was superior in preventing transepidermal water loss.

Another interesting application of liposomal delivery was the preparation of vesicles loaded with Myrtus Communis berry extract [24], renowned for its rich antioxidant profile, confirmed by HPLC-DAD analysis (Figure 5) revealing high concentrations of anthocyanins, galloyl derivatives, and flavonoids in the macerated ethanolic extract (1:1 *w*/*v*). Employing a straightforward sonication method with fat-free soybean phospholipids (70% phosphatidylcholine) in PBS, liposomes with a mean diameter (MD) of 102 ± 5.6 nm were produced. These liposomes, while slightly larger than empty vesicles, exhibited comparable polydispersity index. The research group determined the entrapment efficiency (EE) of each main antioxidant of the myrtle extract: the EE ranges from 71.4% of Myricetin-3-O-galactoside to over 95% of most of the anthocyanins. The antioxidant activity of the loaded liposomes, measured by DPPH analysis, was only slightly lower (326 ± 17 μg Trolox Equivalent/mL) than that of the free myrtle extract. The reducing power remained unchanged, as confirmed by FRAP analysis. Furthermore, the 3-(4,5-Dimethylthiazol-2-yl)-2,5-diphenyltetrazolium bromide (MTT) assay on cell viability demonstrated the low cell cytotoxicity of myrtle extracts and the even higher vitality of cells treated with loaded liposomes vs. control cells. Moreover, in fibroblasts exposed to the AAPH radical generator (i.e., anti-ROS activity experiments), liposomes loaded with increasing quantities of myrtle extracts (from 0.1 to 10 μg) significatively mitigated the oxidative stress, even better than the pure myrtle extract after 24 h. Unfortunately, no direct tests on skin were performed by authors even with these encouraging results.

Permana, Rahman et al. [25] investigated the use of liposomes to enhance the antioxidant activity, permeability, and skin retention of propolis extracts (PE). The more efficient extraction (75% *v*/*v* of ethanol/water, PE75) in terms of total polyphenolic content (TPC 179.3 ± 9.3 mg/g PE) and total flavonoid concentration (TFC 277.2 ± 3.4 mg/g PE) turned out to be the more antioxidant, either as radical scavenger capacity (DPPH IC_50_ 43.2 ± 3.0) or as lipid peroxidation inhibition (LP, 65.3 ± 3.4 μg/mL). These PE are rich in caffeic acid (maximum 4.15 ± 0.01 mg/g in PE75), quercetin (0.72 ± 0.01 mg/g in PE75), and kaempferol (2.62 ± 0.01 in PE75), as confirmed by HPLC analyses. 1 g of PE was encapsulated into liposomal vesicles using different amounts (1–3 g) of phosphatidylcholine and 0.5 g of cholesterol. The preparation applied the thin-film hydration technique with subsequent solvent evaporation. Higher concentrations of phospholipids, increasing the viscosity of the solution, generally produced larger vesicles due to the higher surface tension. On the other hand, the use of a lower weight ratio of PE/phosphatidylcholine combined with a longer duration of homogenization or bath sonication allowed for a reduction in liposomal size from 620 ± 54 to 248 ± 22 nm in diameter, while PDI and ZP remained constant. Antioxidant entrapment efficiency (EE) correlated strongly with the ability of each molecule to form hydrogen bonds with the liposomal bilayer, acting either as a hydrogen bond donor or acceptor. Quercetin demonstrated superior encapsulation (77 ± 6%), exceeding that of kaempferol (45 ± 4%) and caffeic acid (55 ± 5%). Liposome size and processing parameters (homogenization and sonication time) also influenced EE. The obtained resulting vesicles enhanced the water solubility of caffeic acid, kaempferol, and quercetin by 5-fold, 11-fold, and 12-fold, respectively. LIP were dissolved in a hydrogel (Carbomer 940 and glycerol in water) in order to analyze their potentiality as skin formulations. Antioxidant analysis confirmed that their antioxidant activity, either as a radical scavenger (DPPH IC_50_ 48 ± 5) or as a lipid peroxidation inhibitor (LP, 68 ± 5 μg/mL), remained similar to that of propolis extract hydrogel. However, studies in vitro on mouse skin revealed that the use of LIP hydrogel enhanced the skin permeation and retention of all antioxidants by 4- to 8-fold compared to the hydrogel containing PEs. This was ascribed to the poor absorption of the free antioxidants into the skin and the opposite effect of the phospholipidic bilayer that facilitates the penetration into the stratum corneum. Further, this hydrogel formulation containing propolis extract also acted as a photoprotector against UVA and UVB aggression, as stated by the measured sun protection factor of 17 ± 2.

The combined use of liposomes and other carriers, such as cyclodextrin (CD), has been designed in pharmaceutical applications to enhance the stability of loaded drugs and modulate their release (Modulatory Liposomal Controlled Release, MCLR) [48].

Gardikis et al. [26] analyzed the use of liposomes and CD to enhance the adsorption of propolis antioxidants and study their biochemical activity in skin. In their research they used micronized propolis extracted (PE) in the presence of CD. The commercial LIP (Pro-Lipo^TM^ Neo) formed by propanediol (75.0% *w*/*w*), lecithin (25.0% *w*/*w*), vitamin E (0.25% *w*/*w*), and sunflower oil (0.15% *w*/*w*), mixed with the CD/propolis extract, produced large micelles, the size of which was reduced through membrane filtration. Using 10% *w*/*w* propolis concentration, the obtained liposomes showed an initial MD of 255 ± 36 nm, decent PDI (0.366 ± 0.051), and good ZP (−21 ± 5 mV). This preparation gave higher EE (84 ± 4%) with respect to previous works, thanks to the presence of CD. The TPC and the antioxidant activity determined by Folin–Ciocalteu’s method and DPPH assay, respectively, evidenced a high polyphenol concentration of 100.0 ± 0.7 mg gallic acid eq/mL and a significant antioxidant activity of 33.7 ± 0.2 mM Trolox equivalent. However, both parameters decreased over time or when vesicles were stored at temperatures higher than 6 °C. The liposomal formulation was also studied in vitro on human skin cells (HaCaT) by biochemical and genetic tests (Figure 6).

LIP enhance skin cell metabolism, increases ATP levels, and protects cells against UVB, thereby reducing photoaging. The upregulation of various genes connected with cell proliferation (AQP3), antiaging (TNFα), cell defense against pathogens (IL4), and immune protection (ITGB2) confirmed the beneficial effect of the propolis and the efficacy of the liposomal carrier in enhancing the absorbance of the active principles into the skin. Finally, a cosmetic formulation containing 1% LIP proved that the loaded liposomes were non-irritating for human skin cells, as tested by MTT on a reconstructed skin model, indicating potential for market application.

Oskoueiana, Karimi, and coworkers explored the potential of nanoliposomes loaded with Owhadi pistachio hull extract (PHE) with the aim of enhancing the stability and applicability of a complex polyphenolic antioxidant mixture (TPC 52 ± g GAE/100 g dry extract) including gallic acid, rutin, quercetin, myricetin, catechin, and pyrogallol [27]. The LIP were formed by mixing a hot water solution of lecithin (2% *w*/*w*) and ethanolic pistachio hull extract (0.2% final concentration) and then sonicating. While the resulting vesicles exhibited desirable characteristics (MD: 96.3 ± 7.2 nm; ZP: −48.0 ± 0.8 mV), entrapment efficiency was modest (EE of 38.4 ± 2.7%). As indicated by in vitro studies (MTT assay), the liposomal formulations maintained high cell viability in murine hepatocytes at extract concentrations up to 50 μg/mL. The antioxidant activity was confirmed by DPPH assay, even if LIP radical scavenging capacity resulted to be lower (IC_50_ 12.6 ± 2.9 μg/L) in comparison with gallic acid reference (IC_50_ 8.8 ± 0.6 μg/L). The LIP also up-regulate the expression of endogenous antioxidant enzyme genes in murine hepatocytes: SOD (+3.2-fold change), CAT (+2.1), GPX (+1.9), and Nrf2 (+2.7), although not to the same extent as gallic acid. Furthermore, the PHE liposomal formulation demonstrated activity against melanogenesis: it inhibited tyrosinase activity in mushroom cells (IC_50_ 21.8 ± 4.3 μg/mL) and reduced melanin production. Additionally, it partially suppressed the expression of the tyrosinase, MTIF, TRP-1, and TRP-2 genes in melanoma cells, suggesting potential applications in skin whitening processes.

Addressing the crucial issue of scalability and the challenging endeavor of industrial production, Caddeo, Pucci, and coworkers [28] analyzed the preparation of vesicles loaded with resveratrol from a commercial liposomal preparation to check the possibility of an easy-to-handle formulation for skin application. The Pronanosome LIPO-N (PLIPO-N) by Nanovex Biotechnologies SL (Llanera, Spain) is a commercial powder of dry LIP. The easy preparation of the liposomal carrier, through an overnight mixing of resveratrol (2 mg/mL), Tween 80 surfactant (10 mg/mL), and PLIPO-N (100 mg) in water, followed by sonication, yielded highly homogeneous nanovesicles (PDI 0.19) having MD (80 nm) and ZP (−25 mV) statistically consistent with the same regenerated empty liposomes. The loading of the resveratrol was high (EE 83%), and long-term tests confirmed the stability of the physicochemical properties of the vesicles for more than three months. The formulation maintained resveratrol’s antioxidant activity (DPPH assay: 321.0 ± 7.0 μg of Trolox equivalents/mL; FRAP assay: 8.1 ± 0.7 mg of Fe^2+^ equivalents/mL) and significantly reduced its cytotoxicity compared to the ethanolic solution, as demonstrated by 3T3-L1 cell viability assays. Indeed, surprisingly, while resveratrol ethanolic solution lowered 3T3-L1 cell viability up to 20% at 10 μg maximum concentration, the loaded LIP proved to be less toxic with a 55% viability in the same condition. The ability of resveratrol to protect fibroblasts from ROS injuries was analyzed by dichlorofluorescein (DCF) assay. In the DCF experiments, the fluorescence lowered as resveratrol concentration increased much faster than with the liposomal formulation. On the other hand, microscope images confirmed that, using free resveratrol, the cell mortality increased so much as to invalidate recorded data. Instead, the ROS scavenging capacity of liposomal resveratrol did not produce cell death at the higher concentration, thus supporting the usefulness of the liposomal formulation.

Surfactants, which are amphiphilic, generally are used to stabilize the vesicles, reduce their diameter, and increase their deformability by lowering the energy required to modify membrane curvature. For this reason, adding surfactants to liposomal formulation could be useful for a dermal formulation.

In the case of naringenin, another antioxidant that is poorly soluble in water but shows high potential as an anti-inflammatory agent and regulator of fibroblast and melanoma cell growth, Lowry et al. [29] investigate its incorporation into liposomal formulations containing different percentages of Tween 20 surfactant. The positioning of naringenin in the bilayer increased the diameter of the vesicles. The so-formed loaded liposomes had a MD of 392 ± 32 nm that decreased to 235 ± 9 nm in the presence of 10% Tween 20. The vesicles resulted neutral or slightly negative (ZP from 4.12 to −0.22 mV) but with a PDI of 0.3%. The EE of the active principle was also decreased by Tween 20, from 90.8 ± 4.6% to 64.3 ± 5.6% at 10% Tween 20 concentration. It was supposed that the surfactant competed with lipophilic naringenin for a position within the membrane. As the authors expected, the elasticity of the different liposomal formulations increased in the presence of a higher percentage of surfactant, as evidenced by a decrease in the deformability index (DI) from 80.7 ± 2.0% (without Tween 20) to 59.2 ± 4.4% (with 10% Tween 20). Gel derived from hydroxyethyl cellulose (HEC) and hydroxypropyl methylcellulose (HPMC) was added to naringenin and to the loaded liposomal formulations to compare the release of the active principle using side-by-side diffusion cell experiments (50 nm pore size membrane). The results showed that the liposomes released naringenin faster than the solution, and the release rate increased as the concentration of Tween 20 reached 10%. On the other hand, the comparison of the formulations containing either loaded liposomes with 2% Tween 20 or without, and one cellulose gel, evidenced that HMPC gel slowed the release of the naringenin better than HEC gel even in the presence of the surfactant that enhanced the process in both formulations. The study of naringenin uptake by HaCat and HDFa cells using the sodium 3′-(1-phenylamino)-carbonyl-3,4-tetrazolium (XTT) assay as a fluorescent dye confirmed the ability of gel liposomes to penetrate the cells, likely due to ionic interactions with the cell membrane. These data illustrate the importance of both the vesicle structure and the formulation employed in the delivery system for achieving a synergistic effect on the transport and adsorption of the active ingredient.

### 2.2. Unsaturated Fatty Acid Liposomes

Unsaturated fatty acid liposomes (UFS, ufasomes), composed of a mixture of salified and non-ionized unsaturated fatty acids, represent an alternative vesicular delivery system. The use of ufasomes has followed a parallel but slower development in recent decades [49,50]. UFS formation is pH-dependent, typically requiring a pH range of 7–9, depending on the acids used, to ensure the presence of both forms and achieve stable vesicles rather than micelles (at higher pH) or oily emulsions (at lower pH). As UFS are vesicles formed by a single lipophilic chain, they are more deformable. Furthermore, there is a dynamic equilibrium between the free acid molecules in the solution and those present in the membrane bilayer. Oleic acid (C18, Δ9) and linoleic acid (C18, Δ9–12), commonly chosen for their biocompatibility and intrinsic biological activity as functional food, also exhibit skin-whitening properties via melanin production inhibition [51]. Paolino et al. [39] investigated UFS containing equal molar quantities of oleic and linoleic acids, with (WL) or without lecithin as an aggregating molecule for the topical delivery of ammonium glycyrrhizinate (AG, 3 mg/mL), a potent antioxidant and anti-inflammatory agent. Both formulations (with and without lecithin) exhibited desirable characteristics: narrow polydispersity indices and negative ZP (from −42 to −50 mV). The MD, higher for empty vesicles or without the presence of lecithin, ranges from 146 ± 1 nm to 153 ± 3 nm in the loaded samples, due to an aggregating effect of the active principle. Both compositions showed a high EE (81–85%). As expected, the deformability index of UFS was good in all formulations, and the presence of the antioxidant enhanced this property. However, in the Franz diffusion cell test, an unexpectedly low release capacity was observed in vesicles containing only unsaturated acids (6.0 ± 0.8%) and in those supplemented with lecithin (15.5 ± 1.0%). The comparison of percutaneous permeation on SC cells between the two UFS formulations and an ethanolic solution of AG demonstrated more clearly the superior permeation capacity of the vesicles. In fact, a permeation rate of 15.0 ± 4.9 μg/cm^2^ h for UFS WL and 16.6 ± 5.2 μg/cm^2^ h for UFS was obtained. In comparison, the permeation rate for the ethanolic solution was only half that of UFS, as AG had limited ability to penetrate the SC due to its large amphiphilic molecular structure. Experiments on NCTC2544 cells with UFS loaded with 0.03 mM AG confirmed the complete safety of both the vesicle formulations. The UFS demonstrated superior protection for cells stressed by H_2_O_2_ compared to the AG solution at the same concentration, resulting in cell viability exceeding 80% for all formulations. Furthermore, these vesicles succeeded in decreasing twofold the release of lactic dehydrogenase in comparison with AG solution. As the unloaded UFS did not exert any antioxidant activity, the enhancement of glycyrrhizinate’s effect should be attributed to the higher internalization of the active ingredient into the cells, facilitated by the delivery system. Finally, skin tolerability testing indicated that UFS did not disrupt skin lipid structure, thus facilitating penetration without causing tissue damage.

### 2.3. Ethosomes and Transethosomes

In recent years the use of vesicles in therapeutic applications has significantly increased, particularly due to the development of liposomal derivatives characterized by enhanced mobility and deformability. These vesicles are particularly effective in dermal treatments. Ethosomes (ETH) are soft, malleable structures composed of phospholipids, ethanol (20–45%), and water, classified as ultra-deformable vesicles [52]. Their unique composition allows them to penetrate deeper skin layers and facilitate greater drug permeation compared to traditional liposomes (LIP). Their flexible structure and ability to enhance skin penetration is largely attributable to ethanol’s ability to perturb the organization of SC lipids. The deformability index (DI) can parametrize the deformability of these vesicles, with a lower DI indicating a higher degree of deformability. This index can be determined by measuring ethosomes’ diameter and weight before and after extrusion through membrane filters with pores smaller than the average vesicle size. The exceptional deformability of ETH make them a valuable tool for transdermal drug delivery in pharmaceutical and cosmetic formulations. Furthermore, ethosomes are particularly advantageous for delivering drugs with poor water solubility or high molecular weight.

Sguizzato and coworkers [30] prepared ETH containing coenzyme Q10 (CoQ10) to enhance the bioavailability of this important antioxidant in the skin. CoQ10 is insoluble in the most common solvents and poorly soluble in ethanol (0.3 mg/mL). The preparation of ETH from a solution of soybeans PC (0.6–1.5% *w*/*w*), CoQ10 (0.1% *w*/*w*), and ethanolic solution (28.5–29.5% *w*/*w*) in water (70% *w*/*w*) was performed at room temperature and enhanced the solubility of the antioxidant to 1 mg/mL, allowing for 98% adsorption into the vesicles. This procedure resulted in better performance if compared to that obtained with solid lipid nanoparticles or simple LIP, which permitted an adsorption of only 73% and 89%, respectively [53]. Regarding the EE parameter, the most stable vesicles obtained were derived from a 0.9% *w*/*w* concentration of PC in ethanol, with a 254–271 nm MD and a mixed unilamellar and multilamellar structure.

Over a three-month period, the stability of CoQ10 inside the ETH was higher than that of the free molecule or other supramolecular preparations. ETH alone or with CoQ10 was found to be safe for cells in MTT and LDH experiments. The research group demonstrated the ability of loaded vesicles to penetrate fibroblast membranes, probably by endocytosis, and protect human fibroblast cells from H_2_O_2_ insults in ex vivo experiments, as demonstrated by analysis of the production of 4-hydroxynonenale (4-HNE) from lipids. The same fluorescence experiments on reconstructed human epidermis and TEM images (Figure 7) confirmed ETH adsorption within the stratum corneum (SC) and even into the stratum basale, as well as their protective effect against hydrogen peroxide-induced lipid degradation, resulting in a 35% reduction in the production of 4-HNE.

To fully harness the powerful antioxidant properties of rutin, given its difficulty to go beyond the SC, the set-up of an effective delivery system is almost essential. Rolim Baby et al. [31] addressed this challenge by formulating ethosomes with PC from egg yolk (2% *w*/*v*), rutin (0.03% *w*/*v*), and a hydroalcoholic solution (20% ethanol). The vesicles produced were larger (369.5 ± 5.2 nm) than the empty one, with a lower ZP (−19.6 ± 0.4 mV) and a higher PDI (0.43 ± 0.01). In particular, the elevated PDI exceeding 0.3 could be due either to the formation of vesicle agglomerates during the addition of the rutin ethanolic solution or to a more disperse geometry of the vesicles. In vivo tape-stripping tests on human skin demonstrated that rutin-loaded ETH were not irritating and that they penetrated the lower layers of the stratum corneum (SC) three times more than the rutin solution. Even if the antioxidant activity in vitro was confirmed for the ETH formulation, the ex vivo DPPH assay, either on the upper or deeper layer of the skin, did not show relevant differences between the loaded ethosomal formulation, the free rutin solution, and the control. The authors ascribed this negative result to the low concentration of rutin in the formulation.

Two years later, Paolino et al. [32] succeeded in enhancing the rutin antioxidant efficacy with respect to the simple rutin solution by studying a different ethosomal composition. The ETH were prepared with Phospholipon 90G^©^, water, and rutin ethanolic solutions, in the dark to avoid rutin photodegradation. The concentration of rutin solutions, from 0.5 to 4 mg/mL, into the ETH was inversely correlated with the size (MD from 110 ± 8 nm to 575 ± 6 nm) and also with the elasticity of the vesicles. However, DI values (from 3.5 to 10%) permitted a reasonable degree of membrane elasticity. Also, the ZP of the different ETH varied as a function of the rutin adsorbed into the bilayer. In all cases, ranging from −22.4 to −29.5 mV, ZP values were consistently negative and low enough to ensure effective repulsion of the vesicles. Interestingly, in comparison with the other formulations, those ETH prepared with a higher concentration of ethanol (40% *w*/*v* instead of 30% *w*/*v*) produced smaller ZP (from −5.8 to −12.8 mV) and were discarded. The EE was dependent on different parameters: a higher concentration of rutin solution significantly enhanced the EE, while a higher ethanol content reduced it. On the other hand, rutin diminished the release of the active principle by enhancing bilayer rigidity. Considering the higher EE (67.5 ± 5.2%) and DI (3.5 ± 0.5%) obtained, the best ethosome composition was found to be 4 mg of rutin dissolved in 30 mL of EtOH, 69 mL of water and 1% *w*/*v* PL90G^©^. The experiments indicated that this delivery system not only stabilized the antioxidant but also significantly reduced photodegradation under UV light exposure. The MTT assay on cell lines subjected to oxidative stress via H_2_O_2_, demonstrated that the ETH not only maintained cell viability but also exerted antioxidant protection on the cells, unlike the pure compound. This protective effect was concentration-dependent and increased with rutin levels up to 100 μM. Finally, the anti-inflammatory effect of rutin ETH, conducted on conveniently irritated epidermis of human volunteers, confirmed the higher effect of the loaded vesicles in comparison with a simple solution of the active principle. It should be noted that rutin properties counterbalanced the initial irritating effect of ethanol on the skin due to the hydroalcoholic preparation, while the erythema recovery, in the presence of rutin ETHs, was more rapid and with better results after 4–5 h of treatment.

In the case of *Euphorbia characias* latex extract, ETH were conveniently used as a skin penetration enhancer of the quercetin glycosides [33]. This extract inhibited collagenase, hyaluronidase, and elastase enzymes responsible for skin aging. Suitable ETH were prepared with soy PC (120 mg) and *E. characias* extracts (2 mg) in ethanol (0.1 mL) and water (0.9 mL). The obtained vesicles were nanostructures (MD 101.1 ± 9.7 nm, PDI 0.29 ± 0.07%) with high ZP (−70.6 ± 7.3 mV) and a good EE of 85–86%, depending on the glycoside bound to quercetin. MTT experiments pointed out the safety of the vesicles, with cell viability > 82% for empty and even higher for loaded ETH, while the DPPH assay confirmed the maintenance of high antioxidant activity of 94.8 ± 0.4% in the liposomal formulation. The anti-melanogenic effect on tyrosinase, assessed by zymography on cells treated with the L-DOPA enzyme activator, confirmed a high level of inhibition by the plant extract, which was further enhanced when the extract was loaded in the ETH. All of these properties, along with the long-term stability of the prepared ETH, confirm their potential for delivering plant extract bioactive compounds through the skin, even though no specific experiments were conducted on skin cells in this case.

Another recent application of nano delivery systems involves the use of two or more components together for a synergistic effect. Ferrara et al. [34] explored this concept by utilizing ETH-containing curcumin (Cur) and piperine (Pip) to mitigate skin damage derived from environmental pollution. Cur is widely recognized for its powerful antioxidant, antibacterial, and antiviral properties, as well as its effects against inflammation and various pathologies [54]. Meanwhile, Pip shares these properties, but it also increases the bioavailability of other molecules [55] as well, by slowing down the cell efflux systems and enhancing skin cell permeation [56]. In their research, Ferrara and colleagues compared two methods for producing ETH: the bulk production method, in which water was added dropwise to a solution of PC (0.9% *w*/*w*) and ethanol (30% *v*/*v*), and a microfluidic technique. In the microfluidic approach, the flow ratio of the two solutions was maintained at 2:1 (*v*/*v*), while varying total flow rates were applied. During flows mixing, the LIP were formed with homogeneous size as a function of the total flow rates. Nevertheless, the bulk technique turned out to be more rapid (5 mL vs. 0.36 mL volume of 200 nm diameter ETH), also considering a possible scaling up of the production. The ETH so prepared were multilamellar and showed an EE of 97 and 79% for Cur and Pip, respectively. In a second step, the researchers compared the antioxidant effect of Cur and Pip solutions and their ETH, either alone or together. The results evidence that the mixed ETH solutions enhance the drug efficacy of Cur in the FRAP assay. However, some drawbacks can be envisaged in the release profile of the two drugs. In comparison with the drug loaded in the vesicles, the percentage of drug released from ETH was 33% in the case of Pip and 15% for Cur, lower than the values obtained by the corresponding drug solutions that were 57% and 24%, respectively. In the permeability tests, Cur and Pip behaved differently. Pip was faster in diffusing through the synthetic membrane (STRAT-M^®^) that simulates the SC, either in solution or in loaded ETH, the second being slower. Conversely, Cur in ETH showed a partition coefficient higher than the Cur solution even if both had a longer lag time than the Pip systems. If compared to those of Pip, Cur, either in solution or in ETH, also showed a low permeability coefficient. Nevertheless, the Cur ETH system seemed to be more efficient than the corresponding solution in providing a prolonged protective effect. Additionally, the combined use of Cur ETH and Pip ETH further enhanced this behavior against oxidative stress in skin submitted to diesel engine exhausts.

Transethosomes (TETH) represent a further variation in ethosomes containing, together with ethanol, a surfactant that acts as an SC permeation enhancer, which is particularly advantageous for dermal drug delivery [57].

To investigate the antioxidant activity of mangiferin, a glucitol xanthone found in many plants and fruits (papaya, mango), Valacchi, Esposito et al., prepared ETH and TETH as nano delivery systems, considering its higher solubility in ethanol (0.20 mg/mL) with respect to water (0.11 mg/mL) [35]. They analyzed the different loaded vesicles in their physicochemical characteristics, antioxidant activity, and anti-inflammatory properties on skin keratinocytes. The ETH were formulated by mixing PC (30% *w*/*w*) and ethanol (30% *v*/*v*) with mangiferin (3.3 mg/mL), enhancing its solubility to 3.3 mg/mL thanks to PC presence. TETH were prepared by adding, to the same ingredients, different quantities of surfactants (0.1–0.6% *w*/*w*), including hydrophilic polysorbate (Twin 20, Twin 80), lipophilic sorbitane monoleate (SP20, SP 80), or ionic dimethyldidodecylammonium bromide (DDAB). The vesicle size and zeta potential varied depending on the surfactant used, while the PDI remained below 0.2, indicating good homogeneity in vesicle volumes. Generally, a higher concentration of surfactant produced a lowering of the diameter and zeta potential. Except for DDAB, which resulted in highly positively charged vesicles, the ZP were always negative. As already stated, stability tests confirmed that low ZP values were detrimental to the stability of the vesicles, leading to their potential fusion over time due to insufficient repulsive force. Consequently, the research group focused their subsequent studies with mangiferin (0.1% *w*/*w*), only on the vesicles containing either Twin 80 (MD 169.3 ± 0.46, ZP −28.29 ± 0.4) or DDAB (MD 86.41 ± 1.09, ZP 84.08 ± 0.6) as surfactant. MTT assay revealed that while ETH combined with Tween 80 exhibited low cytotoxicity that increased with mangiferin concentration, both empty or loaded DDAB ethosomes showed high toxicity and were discarded from further experiments. Such a result is likely due to the destabilizing effects of the positively charged vesicles on keratinocytes, leading to their lysis. Diffusion tests, conducted with a Franz cell using a nylon membrane, confirmed a lower diffusion capacity for both formulations compared to a simple mangiferin solution; however, TETH showed better performance compared to ETH, because the surfactant most likely facilitated the release of the active principle from the vesicles. Finally, exposure to cigarette smoke was employed to assess the antioxidant and anti-inflammatory effects of the vesicles on HaCaT cells. The results demonstrated that both ETH and TETH permitted mangiferin to penetrate cells (TEM images) and to lower the inflammation of exposed keratinocytes as further demonstrated by the reduction in OH-1 and IL-6 inflammatory enzyme expression.

Subsequently, the same research group investigated the anti-melanoma activity of different concentrations of pure quercetin loaded into either ETH or TETH [36]. Building on previous findings, Tween 80 was employed as the only surfactant. Both vesicle types were multilamellar with similar diameters dependent on the PC and surfactant concentration but with generally uniform dispersity index. Quercetin, incorporated in the bilayer, produced a loosening of the membrane and an enlargement of vesicle diameter while Twin 80 enhances vesicle stability (ETH MD 258 ± 21 nm, TETH MD 240 ± 21 nm).

The entrapment capacity of TETH was slightly higher than that of ETH, with a percentage of 59.2% and 56.4%, respectively, while antioxidant capacity on lipid proved to be almost equal for the two delivery systems (3.16 μmol TE/g vs. 3.26 μmol TE/g) but lower than free quercetin solution. In vitro release tests (IVRT) conducted on lipophilic polytetrafluoroethylene (PTFE) membranes, which closely resembled human skin, exhibited a constant release rate over time (zero-order kinetics), confirming the ability of the vesicles to control and prolong the release of the active principle. In vitro permeation tests (IVPT) confirmed that transethosomes possessed a better capacity in letting quercetin penetrate into the SC with respect to ETH (4-fold lower) and quercetin solution (10-fold lower) (Figure 8).

The MTT experiments, conducted on both HT144 melanoma and HaCaT cells, demonstrated low cytotoxicity at quercetin concentrations below 5 µg/mL. Surprisingly, the subsequent wound healing assay, performed at 2.5 μg/mL quercetin concentration, evidenced a lower capacity of TETH and ETH to favor wound resolution in both the cell lines. This has been ascribed to the capacity of the active principle to inhibit ROS production, thus lowering the natural response of cells to repair the tissue damage, partially activated by ROS as a second messenger.

### 2.4. Niosomes

A different improvement in the use of synthetic vesicles is the development of nonionic surfactant bilayers, including alkyl ethers, alkylated sugars, and polyethylene fatty acid esters, known as niosomes (NIS). Niosomes exhibit higher chemical stability compared to LIP and allow for improved control over shape and size, prolonged drug release, and enhanced bioavailability, making them a preferred option in many delivery systems. Additionally, the cost-effectiveness and scalability of NIS production contribute to their advantages over conventional LIP [58,59].

Al Saqr and coworkers [37] prepared NIS to study the skin adsorption of hydroxytyrosol, an antioxidant found in olive oil that offers various benefits, including anti-inflammatory, anti-platelet aggregation, and anti-UV-B DNA damage properties, among others [60]. Different types of vesicles were prepared by varying the concentrations of ingredients, including Span 60 surfactant (27.6–45%), lecithin (0–15%), cholesterol (5% *w*/*w*), and hydroxytyrosol (1% *w*/*w*). The vesicles were formed by adding buffered water (0–9%) to a hot solution of these components diluted in ethanol (40–45%). Span 60 was chosen as a surfactant for its higher transition temperature (53–57 °C) and its high EE. The use of different percentages of lecithin and surfactant resulted in vesicles with similar small diameters (MD around 1 nm), high negative charge (ZP between −41.2 and −56.4 mV), and high entrapment efficiency (all EE > 92%). In the TEM images, the niosomes utilized in the gel formulation appeared as non-aggregated, spherical-shaped vesicles. The release capacity of niosomes was assessed and compared with ethanolic and PBS aqueous solutions of hydroxytyrosol. In vitro results confirmed the slower release properties of the niosomes, which were associated with the Span 60/water concentration ratio and the surfactant’s ability to incorporate hydroxytyrosol into the bilayer. In vivo permeation and retention experiments were conducted using human cadaver skin. The results confirmed that niosomes had a greater ability to transport hydroxytyrosol through the SC and facilitate its penetration into skin cells compared to antioxidant solutions. The authors attributed this effect to the capacity of Span 60 to deconstruct the SC. The concentration of water used in NIS preparation also played a significant role, as evidenced by the lower transdermal permeation and retention of hydroxytyrosol from niosomes prepared without water. Analogously, a higher concentration of lecithin lowered the delivery of the active principle.

The group led by Jingyuan Wen investigated the optimal niosome formulation for delivering epigallocatechin gallate (EGCG) into skin cells using mathematical methodologies together with statistical optimization techniques [38]. They examined the effects of the concentration of EGCG, surfactant (either Twin 40 or Span 60), and dihexadecyl phosphate (DCP), the cholesterol/surfactant ratio, the buffer amount (PBS solution with 10% EtOH, 7.4 pH), and hydration time on encapsulation efficiency. Different batches of niosomes were prepared, and their encapsulation efficiency (EE) was measured. A multivariable equation was developed in which EE is the dependent variable. ANOVA statistics confirmed the significance of both the amount of antioxidant and the cholesterol/surfactant ratio in achieving higher EE. The subsequent experimental data confirmed that niosomes containing 1.4 mg of EGCG and with a cholesterol/surfactant ratio of 0.9 exhibited the highest EE (53.05 ± 4.46%). These niosomes were spherical, as shown in the TEM images, and had a MD of 235.4 ± 15.6 nm, a negative ZP of −45.20 ± 0.03 mV, and a good PDI of 0.267 ± 0.053. Studies on the release of EGCG from loaded NIS, using a Franz diffusion cell, evidenced a longer release time (35% after 3 h, 73% after 21 h) compared to the EGCG solution, which released 100% in 2 h. Additionally, the deposition of EGCG from NIS onto the skin was significantly greater at both 12 (69.0 ± 13.87 µg/cm^2^) and 24 h (54.38 ± 8.86 µg/cm^2^), proving to be twice as effective compared to the EGCG solution. Experiments with fluorescein 5(6)-isothiocyanate (FITC), either free or loaded in the same NIS, demonstrated that the molecule penetrated deeper into the human SC when released from the vesicles rather than from the solution.

The antioxidant activity of EGCG-loaded NIS was evaluated by measuring malondialdehyde (MDA) production in UV-stressed fetal bovine cells. Cells treated with EGCG-loaded NIS exhibited MDA levels that were 38.4% lower compared to those treated with EGCG solution. This effect was attributed to the activation of glutathione peroxidase (GPX) production due to the release of EGCG in the treated cells, confirming the enhanced protective effect of the loaded antioxidant. Notably, the cellular uptake of NIS requires energy, suggesting a time- and concentration-dependent mechanism mediated by endocytosis.

## 3. Antioxidant Delivery System Comparison

First-generation liposomes suffer from physical instability over time, as they tend to aggregate into larger vesicles. New formulations with higher zeta potential (ZP) and surfactants have been developed to address this issue. Furthermore, many phospholipid nanoparticles exhibit low viscosity, necessitating their use in gel formulations for topical applications, in contrast to solid lipid nanoparticles. However, LIP, ETH, and their modifications exhibit a high soothing effect, facilitate easy fusion with keratinocyte membranes, and allow for rapid permeation and retention. Evaluating the optimal drug delivery system for a specific antioxidant remains challenging due to the need to compare diverse systems. Furthermore, areas such as cosmetics, skincare, and skin medicine have distinct needs and require different analytical techniques, which complicate direct comparisons. Therefore, this chapter focuses on the two most studied antioxidants: curcumin and quercetin. It examines several relevant parameters for assessing the performance and effectiveness of each delivery system, including entrapment efficiency, release profiles, efficacy, and compatibility with the skin.

### 3.1. Curcumin

Curcumin possesses numerous biological properties [54] and has low water solubility, making it an ideal candidate for studying new techniques to enhance its bioavailability and adsorption. Consequently, among the antioxidant delivery systems, those incorporating curcumin have been the most extensively studied [61,62,63].

Focusing on lipid nanovesicles, in addition to the previously presented ethosomes [34], solid lipid nanoparticles (SLN, Figure 9) containing curcumin and turmerone have recently been developed by Aydin et al. to alleviate skin soreness [64] (Table 2).

SLN are vesicles generally formed from a mixture of glycerides. Their particle size ranges from 40 to 1000 nm, and their higher capacity for lipophilic drug loading makes them a viable alternative to liposomal and ethosomal formulations. As they penetrate the stratum corneum and accumulate in the epidermis and dermis, SLN are often used to deliver drugs that are effective in treating psoriasis and other epidermal imbalances. In the case of curcumin, SLN were produced using curcumin (0.02% *w*/*v*) and turmerone (0.1% *w*/*v*), both dissolved in behenic acid glycerides and mixed with a nonionic surfactant. The computer-aided study, validated by experimental data, achieved encapsulation efficiency of 77% for curcumin and 75% for turmerone. Surprisingly, these data revealed a lower EE compared to that obtained with ethosomes. The release of curcumin and turmerone from SLN was initially rapid: 48.2 ± 2.3% of curcumin and 54.2 ± 4.5% of turmerone were released within the first 2 h. The remaining antioxidants were released more slowly: after 24 h, a total of 71.3 ± 3.7% of curcumin and 67.2 ± 1.6% of turmerone had passed through the dialysis membrane used for measurement. The biphasic release of lipophilic compounds is a characteristic feature of SLN. The initial rapid release can be attributed to the molecules associated with the surface of the particles, while the subsequent slowdown was due to the diffusion of molecules more deeply embedded within the nanoparticles. Unfortunately, the dialysis using a cellulose membrane did not provide information on retention within the epidermis and dermis, which would have enabled a complete comparison with the data obtained from ethosomes.

Nanostructured lipid carriers (NLC, Figure 8) are another type of delivery system used for curcumin. They are composed of solid and liquid lipids, which create a blended phase. The group of Rodriguez-Ruiz et al. [65]. produced NLC by mixing the lipid phase, composed of Precirol^®^ ATO 5, Labrafac, Tween 80, and curcumin (2.4% *w*/*w*), with a water phase containing a surfactant at 70 °C. The resulting lipid nanoparticles exhibited a bimodal distribution. The encapsulation efficiency (EE) was 83–84% and 55% after size exclusion chromatography, which was employed to completely purify the lipid carriers from any free curcumin outside the particles. Like solid lipid nanoparticles, the lipophilicity of NLC facilitates the absorption of substances both within and outside the nanoparticles, making it challenging to achieve complete control over antioxidant release. The lipid nanoparticles, both alone and incorporated into a gel, were compared regarding their permeability and retention. The resulting data demonstrated an exponential penetration of curcumin into the skin over a 24 h period. In parallel, no curcumin was detected leaving the Strat-M membrane, confirming its complete retention within the synthetic membrane that mimics the skin.

The three systems analyzed demonstrate significant differences in the delivery of curcumin. SLN appear to favor skin permeation, while NLC exhibit the opposite behavior, showing good retention within the skin. Ethosomes display an intermediate behavior, as they partially penetrate the membrane while also maintaining some retention. The varying partitioning of these delivery systems through the skin can be advantageous for targeted applications. For drugs that need to remain entrapped in the epidermis and dermis the NLC or ethosomes should be preferred. Conversely, for drugs intended for transdermal absorption, the SLN are the most suitable choice.

### 3.2. Quercetin

Many delivery systems have been employed to enhance quercetin absorption [66]. In the field of lipid nanoparticles, which this review seeks to investigate, quercetin has been delivered not only using liposomes [23], ethosomes, and transethosomes [36], but also through solid lipid nanoparticles (SLN) to reduce skin permeation and water loss [67] as well as natural lipid nanoparticles (NLN) [68] (Table 3).

In the SLN formulation, palmitic acid (5 *w*/*v*%) was combined with a varying percentage of Tween 80 (0.5–4 *w*/*v*%) to load quercetin (0.5 *w*/*v*). The SLN demonstrated EE that depended on the percentage of Tween 80 used, with the highest value of 46.2 ± 2.2% observed in the formulation containing 2% surfactant. These efficiencies, however, are lower than those of liposomes (65–68%), ethosomes (56.4%), or transethosomes (59.4%) (Table 3). In in vitro permeation tests (IVPT), SLN demonstrated a higher permeability (33.5 μg/cm^2^ or 21.9%) compared to all the other vesicles, although the different membranes used added variability to the experiment comparison. Conversely, in vitro retention tests (IVRT) yielded different results, with transethosomes exhibiting the highest retention values (54–64%). The retention of SLNs in the epidermis and dermis was only 2.2% of the total amount, while the highest quantity that passed through the skin was 18.7%.

In the NLN delivery system developed by Marcato et al., illipe butter and calendula oil were utilized to solubilize quercetin [68]. The lipid nanoparticles were produced by adding water, mixed with detergent, to the lipid/quercetin mixture at 50°C under sonication. The lipophilic quercetin was effectively encapsulated within the NLN, achieving an EE of 97–98%, which is higher than that of all previously reported systems. Additionally, the low crystallinity of the lipids employed contributed to the high stability of the delivery system, which maintained its integrity for 90 days without a significant loss of quercetin. The experiments confirmed that the antioxidant activity of the NLN was comparable to that of quercetin in solution. In in vitro permeability and retention tests, these lipid nanoparticles allowed for complete penetration of the formulation into the SC with an 86% retention in the epidermis and dermis while preventing permeation beyond the skin. These properties help avoid systemic drug release, which is particularly useful when the active principle needs to function within the skin, such as in the development of photoprotective systems that must not penetrate beyond the skin to prevent potentially dangerous systemic effects.

The last two examples illustrate how the appropriate choice of delivery system can influence the final destination of antioxidants. Systems designed for transdermal delivery require higher penetration and lower retention capabilities. Conversely, a delivery system that is less penetrating can increase the concentration of the active ingredient in the deeper layers of the skin, as happens with ethosomes and transethosomes, and even more with natural lipid nanoparticles, which exhibit good retention.

## 4. Lipid Nanoparticles in Cosmetic Formulations

The cosmetics industry is experiencing rapid growth, with nanotechnology playing a significant role in product development. Among the most notable innovations are nanoscale materials that exploit unique physicochemical properties to enhance the stability, efficacy, and delivery of active ingredients across the skin barrier. Lipid nanoparticles, particularly liposomes, serve a dual function in cosmeceutical formulations: they act both as vehicles for active ingredients and as active ingredients themselves [69]. When used as a delivery system in the cosmetics industry, liposomes offer a multifunctional role: they enhance the penetration, solubility, and stability of active ingredients; prolong the effects of the loaded compounds; protect them from environmental degradation; target active ingredients to specific sites of action; reduce toxicity; and improve control over pharmacokinetics and pharmacodynamics. In addition, liposomes can help make products more cost-effective [69]. Rahimpour et al. [70] reported the successful application of liposomes in a range of cosmetic products, including sunscreens, beauty creams, anti-aging formulations, and hair care products. Nanoliposomes have also been employed in fragrance delivery systems, such as antiperspirants, deodorants, and lipsticks [71]. Studies have demonstrated that the encapsulation of phenolic compounds, such as epigallocatechin-3-gallate (EGCG), in liposomal carriers can enhance their antioxidant activity and provide controlled release over time [72]. In vivo research, such as that conducted by Liu et al., has shown that loading liposomes with compounds like syringic acid can improve their bioavailability [73]. Several commercially available products utilize the liposomal encapsulation of phenolic compounds, particularly in skin care (Table 4).

For example, Capilene^®^ is a liposome-based cream that encapsulates concentrated herbal plant extracts in a jelly-like liposomal vehicle, integrating omega-3, omega-6, omega-7, and omega-9 fatty acids, as well as ceramide precursors [74]. Sesderma^®^, a well-known brand, uses apple polyphenol extracts or quercetin in a liposomal formulation to treat photoaging, dehydration, wrinkles, and skin spots. Other brands, such as Decorté^®^, Mythos^®^, and Apivita^®^, also incorporate anthocyanins in their liposomal skin treatments [75]. These products have been shown to improve the skin’s lipid layer, with liposomal formulations actively delivering key ingredients to targeted sites for enhanced skin repair. In addition to these benefits, liposomal formulations, such as niosomes, have demonstrated the potential to reversibly reduce the resistance of the skin’s outer layer, facilitating the controlled delivery of active ingredients with reduced toxicity [74]. This makes liposomal formulations particularly useful for products focused on skin soothing and tanning. Despite their promising advantages, liposomal formulations still face challenges in the cosmetics industry. Issues such as low drug loading capacity, inconsistent reproducibility, and physicochemical instability have limited their broader commercial application. However, ongoing research into these challenges continues to drive innovation in the field.

## 5. Conclusions

Poorly soluble or unstable antioxidants represent an unexploited resource that could be extremely useful if an appropriate delivery system is developed to facilitate their broader application. In recent decades, research has led to the development of various nanodelivery systems. Liposomes, along with their modifications (ufasomes, ethosomes, transethosomes, and niosomes), have partially responded to the demand for a reliable, stable, and effective delivery system. It is evident that, at this stage of research, it is insufficient to solely focus on the properties of the materials used to prepare suitable liposomes. Each lipophilic antioxidant can interact with and alter the characteristics of the bilayer. However, research conducted in recent years showed several correlations between the composition of the lipid nanoparticles and the delivery system properties, which can facilitate future development. This overview has highlighted the main questions that a researcher must address when designing a liposomal delivery system, as well as the parameters (MD, ZP, PI, and EE) that should be considered before and during the research. In particular, if the antioxidant drug is lipophilic, it dissolves in the bilayer and can modify the stability of the membrane, inducing changes in tension or curvature that can either increase or decrease the size of the liposome, thereby varying the MD values. On the other hand, drug loading generally has less effect on the polydispersity index (PDI) and Z potential of the liposomal structure. The PDI is more closely related to the components used (phospholipids, surfactant, cholesterol, ethanol), the preparation procedure, and any subsequent manipulation (e.g., extrusion, sonication) that leads to a more defined distribution of vesicle diameters, resulting in a smaller PDI. The membrane charge, measured by ZP, depends on the charge of the phospholipids used and any other charged molecules present in the preparation. In the context of skin, negative zeta potentials have been shown to be more advantageous.

Focusing on transdermal delivery, liposomes have been surpassed by more malleable vesicles such as ethosomes, transethosomes, niosomes, and ufasomes. The use of ethanol allows for a higher drug loading capacity and a more deformable structure, while surfactants enhance membrane stability and provide a high deformability, which is suitable for passing through the SC in transdermal applications. The comparison of different delivery systems was conducted specifically on curcumin and quercetin. A final presentation on the lipid nanoparticles used in the cosmetic industry highlighted the significant impact of this research on the cosmetic market.

## Figures and Tables

**Figure 1 antioxidants-13-01516-f001:**
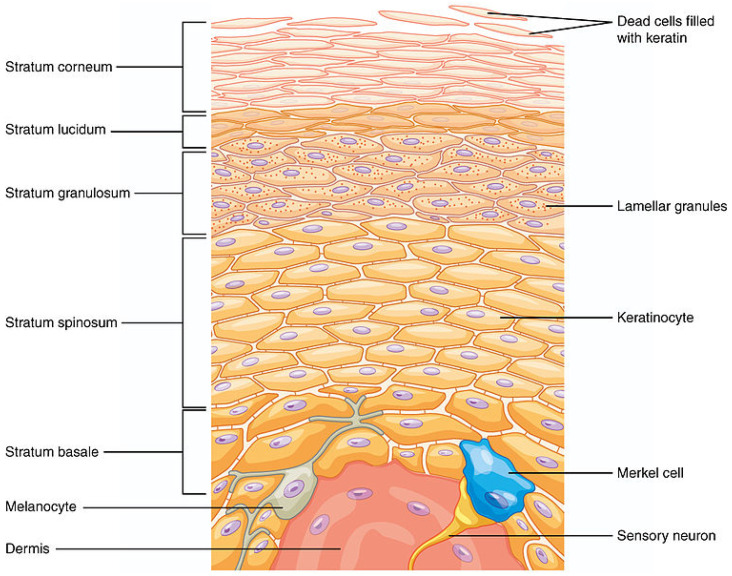
Structure of layers of epidermis (https://commons.wikimedia.org/wiki/File:502_Layers_of_epidermis.jpg (accessed on 16 October 2024)).

**Figure 2 antioxidants-13-01516-f002:**
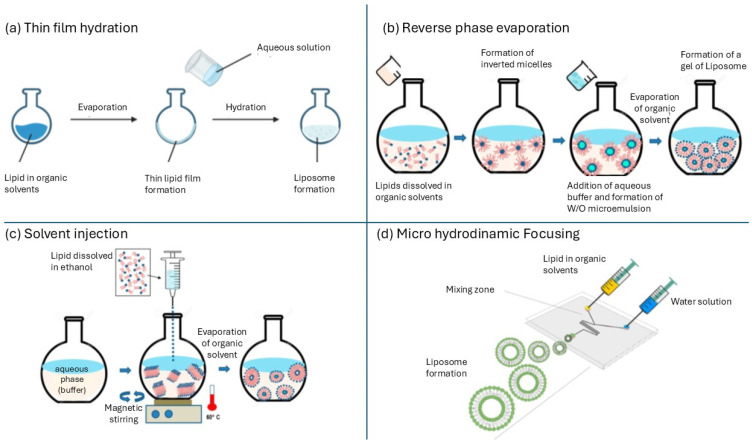
Principal techniques for producing liposomes [42].

**Figure 3 antioxidants-13-01516-f003:**
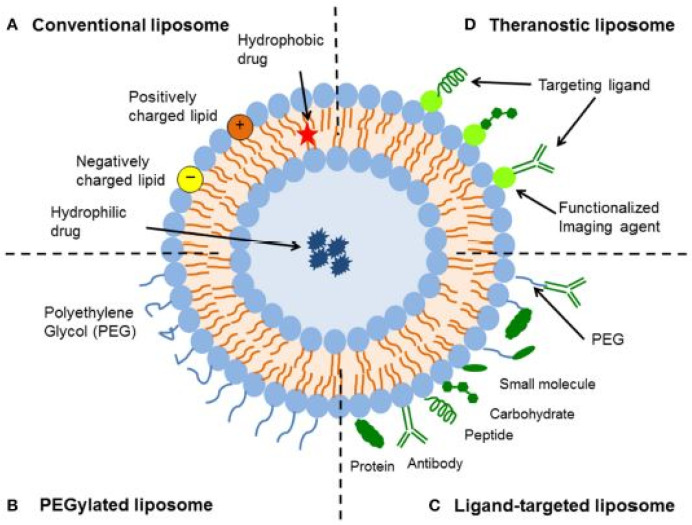
Schematic representation of the different types of liposomal drug delivery systems (https://commons.wikimedia.org/wiki/File:Liposomas_articulo_liberaci%C3%B3n_activos.jpg (accessed on 16 October 2024)).

**Figure 4 antioxidants-13-01516-f004:**
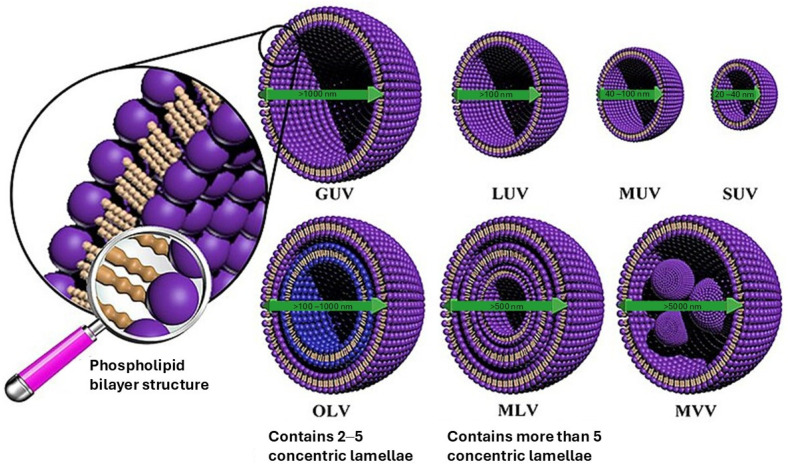
Different sizes and lamellarity of liposomes. V: vesicles, U: unilamellar; G: giant, L: large; M: medium; S: small; OLV: oligolamellar vesicles; MLV: multilamellar vesicles; MVV: multi vesicular vesicles (https://commons.wikimedia.org/wiki/File:1-s2.0-S0168365921005034-gr6_lrg.jpg#filelinks (accessed on 16 October 2024)).

**Figure 5 antioxidants-13-01516-f005:**
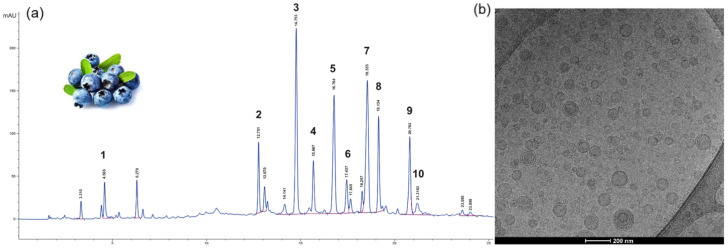
(**a**) HPLC–DAD chromatogram of myrtle berry extract at λ = 280 nm. Chromatographic conditions are described in the reference article. 1: gallic acid; 2: gallic acid derivative; 3: Delphinidin-3-O-glucoside; 4: Cyanidin-3-O-glucoside; 5: Petunidin-3-O-glucoside; 6: Peonidin-3-O-glucoside; 7: Malvidin-3-O-glucoside; 8: Myricetin-3-O-galactoside; 9: Myricetin-3-O-rhamnoside; and 10: ellagic acid. (**b**) Myrtle liposomes through cryo-TEM observation [24].

**Figure 6 antioxidants-13-01516-f006:**
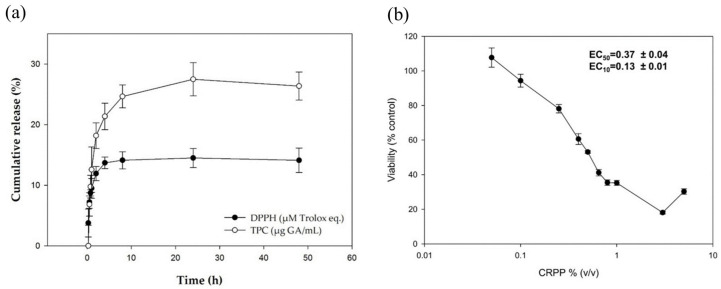
(**a**) Release of antioxidant activity (DPPH) and total phenolic compounds (TPC) of propolis extract (PE) loaded in LIP during 48 h. (**b**) Cytotoxicity profile of PE loaded in human immortalized keratinocyte (HaCaT) cells. HaCaT cells were incubated for 72 h with increasing concentrations (0–5%) of PE loaded in LIP, and their cell viability was estimated via sulforhodamine B (SRB) assay. The EC50 and EC10 values (efficient concentration that causes 50% and 10% decrease in cell viability, respectively) of PE loaded in LIP were determined from the dose–response curves. The results are shown as the mean ± SD of three independent experiments [26].

**Figure 7 antioxidants-13-01516-f007:**
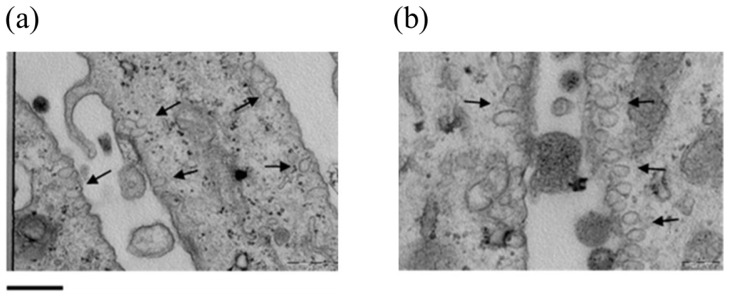
TEM images of fibroblasts in cells treated with ETH (0.9 concentration of PC in ethanol) (**a**,**b**). Panel (**a**), 31.5K magnification; panel (**b**), 50K magnification. ETH are indicated by arrows. Lower bar corresponds to 500 nm [30].

**Figure 8 antioxidants-13-01516-f008:**
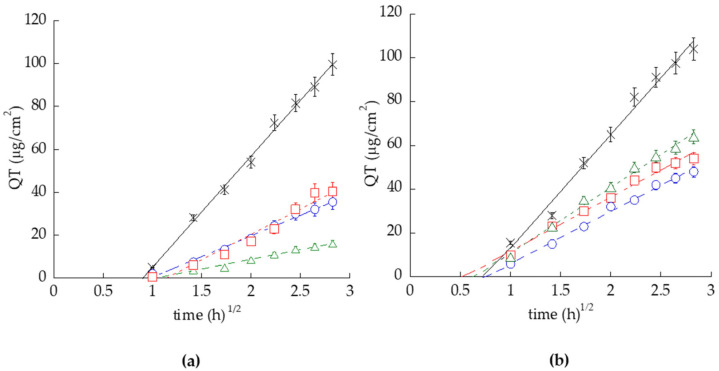
Quercetin (QT) release kinetics from ETH_0.9_-QT (blue circles), TETH_0.9_-QT (red squares), TETH_2.7_-QT (green triangles), and SOL-QT (black crosses), as determined by Franz cell associated with NY (**a**) and PTFE (**b**). Data are the mean of 6 independent experiments ± s.d.

**Figure 9 antioxidants-13-01516-f009:**
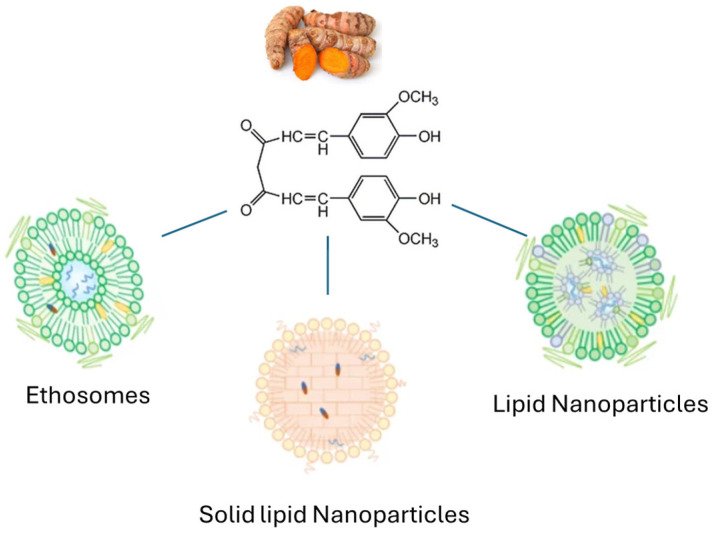
Curcumin lipid nanoparticle delivery systems analyzed.

**Table 1 antioxidants-13-01516-t001:** List of antioxidants analyzed with the liposomal delivery system used.

Antioxidant	Formulation	Delivery System ^1^	Purpose	Ref.
*Moringa oleifera*	extract, cholesterol	LIP	hydration, anti-aging	[13]
quercetin	pure	LIP	skin pressure ulcer	[23]
Myrtus communis	extract	LIP	skin antioxidant	[24]
propolis	extract	LIP	antioxidant, photoprotection	[25]
propolis	extract, cyclodextrins	LIP	antioxidant, anti-aging	[26]
onhadi pistachio	extract	LIP	antioxidant, anti-inflammatory	[27]
resveratrol	pure	LIP	antioxidant	[28]
naringenin	pure, Tween 20	TLIP	antioxidant, anti-inflammatory	[29]
coenzime Q10	pure	ETH	antioxidant	[30]
rutin	pure	ETH	antioxidant	[31]
rutin	pure	ETH	antioxidant	[32]
euphorbia characias	extract	ETH	photoprotective, enzyme inhibition	[33]
curcumin, piperine	pure	ETH	skin antioxidant	[34]
mangiferin	pure	ETH, TETH	skin antioxidant	[35]
quercetin	pure	ETH, TETH	anti-melanoma activity	[36]
hydroxytirosol	pure	NIS	antioxidant	[37]
epigallocatechin gallate	pure	NIS	antioxidant	[38]
glicirrate	pure	UFS	antioxidant	[39]

^1^ LIP: liposomes; ETH: ethosomes; TETH: transethosomes; NIS: niosomes; UFS: unsaturated fatty acids liposomes.

**Table 2 antioxidants-13-01516-t002:** Curcumin delivery systems by lipid nanoparticles.

Compound (%)	Delivery System	EE (%)	Permeation (μg/cm^2^/24 h)	Skin Model	Retention (%)	Skin Model	Application	Ref.
curcumin (0.025%)	ETH	97	24.7 ± 4.1	Strat-M	15.9 ± 2.5	PTFE	skin damage antioxidant prevention	[34]
piperine (0.025%)		79	38.4 ± 5.7		27.9 ± 6.0			
curcumin (0.02% *w*/*v*)	SLN	77	71.3 ± 3.7%	dialysis on cellulose membrane	/	/	skin irritation	[64]
turmerone (0.1% *w*/*v*)		75	67.2 ± 1.6%		/	/		
curcumin (2.4% *w*/*w*)	NLC	83–84 (55)	/	Strat-M	26.9 ± 1.9 22% (24 h)	Strat-M	Antioxidant activity	[65]
	NLC + gel	/	/		6.3 ± 0.7 5% (24 h)			

**Table 3 antioxidants-13-01516-t003:** Quercetin delivery systems by lipid nanoparticles.

Compound	Delivery System	EE (%)	Permeation	Skin Model	Retention (%)	Skin Model	Application	Ref.
quercetin (0.01–0.04 g/mL)	LIP	65–78	5.2–6.8%	mice skin	46–53	mice skin	wound healing	[23]
quercetin (0.5 mg/mL)	ETH	56	1 μg/cm^2^	Strat-M	35.4	NY	melanoma cell migration	[36]
			3 μg/cm^2^	HSCE	48	PTFE		
	TETH	59–64	2.2–1.8 μg/cm^2^	Strat-M	16.1–40.4	NY		
			6–8 μg/cm^2^	HSCE	54–64	PTFE		
quercetin (0.5 *w*/*v*%)	SLN	15–46	33.5 μg/cm^2^ (21.9%)	ICR mice skin	2.20	ICR mice skin	occlusion, water retention	[67]
quercetin (12 mg)	NLN	97–98	not detected	pig ear skin	1.3 μg/cm^2^ (86%)	pig ear skin	antioxidant, antiallergic	[68]

**Table 4 antioxidants-13-01516-t004:** Lipid nanoparticles in cosmetic formulation.

Commercial Brand	Encapsulated Compounds	Nanoparticles Size	Application
Sesderma	Ferulic Acid (apple extract)	80–120 nm	Photoaging, dehydration, wrinkles, and skin spots.
Sesderma	Resveratrol, Quercetin, and EGCG (red grape extract)	80–120 nm	Aging, wrinkles, filmogenic effect, and dehydration.
Decortè	Anthocyanins (purple rice extract)	1–100 nm	Dehydration.
Mythos	Anthocyanins (pomegranate extract)	50–500 nm	Skin repair and protection.
Apivita	Anthocyanins (royal jelly extract)	70–100 nm	Skin regeneration, wrinkles, and increases skin elasticity and antioxidant.
Capilene	Fatty acids (sea-buckthorn–seaberry oil and rice bran oil)	50–500 nm	Skin repair.

## Data Availability

No new data were created.

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
