# Peer review of "Lipid Nanovesicles for Antioxidant Delivery in Skin: Liposomes, Ufasomes, Ethosomes, and Niosomes"

_antioxidants, 2024, doi:10.3390/antiox13121516_

Round 1

Reviewer 1 Report

It is an interasting review about the lipid-based nanosystems for skin applications.

My comments are:

1.The Critical Quality Attributes of lipid-based nanosystems for skin applications.

2. Marketed products.

3. Preparation techniques should be described.

4. Mixed systems with lipids should be described.

1.The Critical Quality Attributes of lipid-based nanosystems for skin applications. (in the introduction)

2. Marketed products shoub be included in the conclusions.

3. Preparation techniques should be described (in the Introduction)

4. Mixed systems with lipids should be described. (a new section)

Author Response

Title: The correct terminology is "lipid-based" since ethosomes, niosomes, and other carriers are discussed.

Thank you for the elucidation. We changed the title as suggested.

Major comments

It is an interesting review about the lipid-based nanosystems for skin applications.

My comments are:

1.The Critical Quality Attributes of lipid-based nanosystems for skin applications.

Thank you for the comment. We added a sentence in the introduction about the drawback of lipid vesicles (line 59-61)

  1. Marketed products.

Thank you for the advice. We added a specific chapter about the subject (4. Lipid nanoparticles in cosmetic formulations).

  1. Preparation techniques should be described.

In paragraph 2.1.1, we provided a brief explanation of the four primary techniques used to prepare liposomes.

  1. Mixed systems with lipids should be described.

In new Chapter 3, which discusses the comparison of delivery systems, we included examples of lipid nanocarriers and solid lipid nanoparticles.

Detail comments

1.The Critical Quality Attributes of lipid-based nanosystems for skin applications. (in the introduction)

Thank you for your comment. We have added a sentence in the introduction (line 67-69) to present the argument that is extensively discussed in the manuscript regarding each specific vesicle presented.

  1. Marketed products should be included in the conclusions.

Thank you for the advice. We added a specific chapter (4. Lipid nanoparticles in cosmetic formulations) about the subject.

  1. Preparation techniques should be described (in the Introduction)

In paragraph 2.1, we provided a brief explanation of the four primary techniques used to prepare liposomes.

  1. Mixed systems with lipids should be described. (a new section).

In Chapter 3, which discusses the comparison of delivery systems, we included examples of lipid nanocarriers and solid lipid nanoparticles.

Reviewer 2 Report

The manuscript entitled “Liposome-Based Antioxidant Delivery Systems for Skin Health” offers an interesting overview of vesicular drug delivery systems including the liposomes, niosimes, transferosome and ethosomes. Authors cover different nanocarriers having different structural component but similar vesicular structure and explain their merits and contributions toward the therapy and delivery. It is well organized literature with comprehensive examples and their potential application. However, certain point still needs improvements and there is a need to address some other systems that are also based on lipids or may have the vesicular structure. The manuscript is very interesting and gather the attention of wide spectrum of audience from the scientific community but need major revisions before accepting the manuscript for the publication. Some points for the revisions are given below:

1.       The title of the manuscript only specifies the liposomes that are lipid based vesicular systems. Therefore, it is imperative to change the title and make it more functional that include all the systems that have been used in this manuscript.

2.       Introduction section should include some information about the vesicular or liposomal drug delivery systems with a summary from inception to current form.

3.       In Table 1, the first term entry should be replaced with the serial number or may remove at all.

4.       The term LPS is misleading, because it is standard abbreviation used for lipopolysaccharide, please consider revision of this term with most suitable term throughout the manuscript.

5.       Authors discussed different factors and different parameters related to characterization and functioning of the liposomes, but they did not summarize them in well-organized manners. There should be some tables indicating the outcomes from different studies.

6.       There is need of some subheadings, images, or illustrations to explain the impact and journey of investigation in different aspects.

7.       Unsaturated fatty acid liposomes should be discussed with the liposomes after the first heading.

8.       There should be a comparative analysis among the different systems for the merits and demerits that indicate the translational importance, that is missing.

In general, the manuscript structure needs multiple corrections.

There are numerous grammatic mistakes, punctuations and typos in the manuscript that require a revision. 

Need further subheadings and arrangements

should add Table and Figures

clearly explain the merits and demerits

explain the structural composition differences

mention the most commonly used components

Author Response

Title: Please consider the revision of liposome with the vesicular drug delivery systems or any other appropriate term, because the manuscript also includes other systems.

Are the timeliness, breadth and accuracy of the discussion qualified?

No. There is need of comprehensive discussion in a comparative manner for different systems.

Thank you for your advice. In the manuscript, during the explanation of the activity of the various systems we already evidenced some differences with other delivery systems. However, as you suggested, we added Chapter 3, where we compare various delivery systems used for the two main antioxidants: curcumin and quercetin.

Is the quality and presentation of the figures satisfactory?

No There is only one Figure in the manuscript. Add other systems and also the applications or functional mechanism through illustrations or figures.

Thank you for your suggestion. We have added various figures (4-8) to the manuscript to enhance readability and presentation of the work.

The English could be improved to more clearly express the research.

We have had the manuscript checked by a native speaker.

Major comments

The manuscript entitled “Liposome-Based Antioxidant Delivery Systems for Skin Health” offers an interesting overview of vesicular drug delivery systems including the liposomes, niosimes, transferosome and ethosomes. Authors cover different nanocarriers having different structural component but similar vesicular structure and explain their merits and contributions toward the therapy and delivery. It is well organized literature with comprehensive examples and their potential application. However, certain point still needs improvements and there is a need to address some other systems that are also based on lipids or may have the vesicular structure. The manuscript is very interesting and gather the attention of wide spectrum of audience from the scientific community but need major revisions before accepting the manuscript for the publication. Some points for the revisions are given below:

  1. The title of the manuscript only specifies the liposomes that are lipid based vesicular systems. Therefore, it is imperative to change the title and make it more functional that include all the systems that have been used in this manuscript.

Thank you for the elucidation. We changed the title as suggested

  1. Introduction section should include some information about the vesicular or liposomal drug delivery systems with a summary from inception to current form.

Thank you for your advice. We have added a sentence in Chapter 2 (lines 87-88) to explain the different inception methods for vesicular formulation. Since our primary focus is on topical administration through the skin, we have also included a reference for those interested in a more in-depth analysis of this topic.

In Table 1, the first term entry should be replaced with the serial number or may remove at all.

We removed the column.

  1. The term LPS is misleading, because it is standard abbreviation used for lipopolysaccharide, please consider revision of this term with most suitable term throughout the manuscript.

Thank you for the advice. We substituted LPS with LIP in all the manuscript.

  1. Authors discussed different factors and different parameters related to characterization and functioning of the liposomes, but they did not summarize them in well-organized manners. There should be some tables indicating the outcomes from different studies.

We appreciate your suggestion. We have added two tables in the new Chapter 3, where a more detailed comparison of the delivery systems is provided.

  1. There is need of some subheadings, images, or illustrations to explain the impact and journey of investigation in different aspects.

Thank you for the suggestion. We added some images (figure 4-8) and tables (2-3) to the manuscript. Furthermore we added two new chapters to the manuscript.

  1. Unsaturated fatty acid liposomes should be discussed with the liposomes after the first heading.

Thank you for your suggestion. We have relocated the chapter on unsaturated fatty acid liposomes (2.2) to follow the chapter on liposomes (2.1)

  1. There should be a comparative analysis among the different systems for the merits and demerits that indicate the translational importance, that is missing.

Thank you for your suggestion. We have added comparison during the presentation of the various systems and expanded the manuscript by adding two new chapters on the comparison of different lipidic delivery systems and the use of lipid vesicles in the cosmeceutical industry.

Detail comments

In general, the manuscript structure needs multiple corrections.

Thank you for your advice. We have enriched the manuscript by adding several sections, including liposome preparation methods, a comparison of various delivery systems used for the same antioxidant (Chapter 3), and cosmetic applications.

There are numerous grammatic mistakes, punctuations and typos in the manuscript that require a revision. 

We have had the manuscript checked by a native speaker.

Need further subheadings and arrangements should add Table and Figures.  

Figures and Tables were added.

clearly explain the merits and demerits

In the new chapter 3 we used to compare different lipid base delivery systems. In the introduction to the chapter we explained merits and drawbacks of the various liposomal delivery systems and compared them with other lipid nanoparticles.

explain the structural composition differences,

the structural composition differences have been traced during the presentation of each delivery system.

mention the most commonly used components.

We added the composition of the vesicles where missing.

Reviewer 3 Report

 Agnese Ricci and co-authors have presented very informative review based on topical delivery of anti-oxidants via liposomes and other vesicles. The manuscript can be accepted after major revisions:

1.     In Figure 1, apart from the basic anatomy, the fate of liposomes into different layers should be given.

2.     Short forms should be avoided (e.g. Table 1-pure comp. )

3.     This review gives very basic information about the vesicles. Therefore, some important points should be introduced to make the manuscript more interesting.

a)     Clinical trial status of liposomes for skin applications (in tabular form).

b)     How composition of liposomes affects the stability and %EE of the active molecules.

c)     What are future prospects of liposomes in the cosmetic industry?

d)     How the stability issues can be solved using liposomes?

e)     What are the scaling up possibilities.

f)      Fate of liposomes after application (topical, transdermal), where they release the anti-oxidant molecules.

g)     How liposomes interact with skin and the permeation of liposomes into deeper layers is very crucial to consider

4. What type of gelling system can be employed for the real applications of liposomes?

Good Luck!

N/ A

Author Response

Is the quality and presentation of the figures satisfactory?

No. I am not familiar with the tool "wikimedia" used to draw figures in this review. Kindly check if they are authentic and/ or authors have taken permissions to use them.

The figures presented are free of copyright if the link is provided in the caption.

The quality of English does not limit my understanding of the research.

Major comments

Agnese Ricci and co-authors have presented very informative review based on topical delivery of anti-oxidants via liposomes and other vesicles. The manuscript can be accepted after major revisions:

  1. In Figure 1, apart from the basic anatomy, the fate of liposomes into different layers should be given.

Figure 1 is only explicative of the skin structure. We have represented the fate of liposomal formulation in the graphical abstract and extensively discussed about the topic in the manuscript

  1. Short forms should be avoided (e.g. Table 1-pure comp. )

Thank you for the advice. We correct the Table.

  1. This review gives very basic information about the vesicles. Therefore, some important points should be introduced to make the manuscript more interesting.

Thank you for the advice. In the introduction of each paragraph we  explained the structrure and composition differences of the various vesicles (liposomes, ethosomes, transethosomes, etc.). Where missing, we added information about the vesicle composition in all the studies presented.

  1. Clinical trial status of liposomes for skin applications (in tabular form).

Thank you for your advice. In the new Chapter 4, we have added a table that outlines the use of lipidic vesicles in the cosmetic industry, specifically focusing on those that are already commercially available.

  1. How composition of liposomes affects the stability and %EE of the active molecules.

Thank you for the advice. In the initial part of Chapter 2.1 (lines 146–160), we have already explained the main modifications made to the liposomal structure to enhance retention and stability. Furthermore, we presented specific examples in Chapter 3 that demonstrate the ability of different lipid nanoparticles to load curcumin and quercetin.

  1. What are future prospects of liposomes in the cosmetic industry?

We added a specific chapter  (4. Lipid nanoparticles in cosmetic formulations) to answer this question.

  1. How the stability issues can be solved using liposomes?

We addressed this question in Chapter 2.1 (lines 146–160) by highlighting the modifications made to the basic structure of liposomes with the addition of cyclodextrins and surfactants. Additionally, we discussed the use of ethosomes and transethosomes as solutions to stability issues.

3) What are the scaling up possibilities.

Apart from the specific case of a commercial liposomal system utilized by one research group (reference 38), we have added a dedicated paragraph to highlight the use of liposomes in the cosmetic industry

  1. Fate of liposomes after application (topical, transdermal), where they release the anti-oxidant molecules.

In general, we have demonstrated the ability of each delivery system analyzed to permeate the skin and be retained in the epidermis and dermis, as reported in the reference article. In particular, in Chapter 3, we used these parameters to compare different delivery systems for curcumin and quercetin.

  1. How liposomes interact with skin and the permeation of liposomes into deeper layers is very crucial to consider.

These information are explained in various part of the article but we added a phrase in the new chapter 4 (line 680-81)

  1. What type of gelling system can be employed for the real applications of liposomes?

Thank you for the question. Industrial applications were beyond the scope of this review manuscript. However, we have included a discussion of the use of gels for in vivo absorption analyses wherever specific experiments are summarized in the article. Additionally, we added a section in Chapter 3 addressing the need for gel systems to prevent the dispersion of liposomes with low viscosity (lines 681-683).

Good Luck!

Thank you

Round 2

Reviewer 3 Report

The paper has been improved and can be accepted for publication.

The paper has been improved and can be accepted for publication.